# Modulation of ADAM17 Levels by Pestiviruses Is Species-Specific

**DOI:** 10.3390/v16101564

**Published:** 2024-10-02

**Authors:** Hann-Wei Chen, Marianne Zaruba, Aroosa Dawood, Stefan Düsterhöft, Benjamin Lamp, Till Ruemenapf, Christiane Riedel

**Affiliations:** 1Institute of Virology, Department of Pathobiology, University of Veterinary Medicine, 1210 Vienna, Austria; hann-wei.chen@vetmeduni.ac.at (H.-W.C.); marianne.zaruba@vetmeduni.ac.at (M.Z.); aroosa.dawood@vetmeduni.ac.at (A.D.); 2Institute for Molecular Pharmacology, RWTH Aachen University, 52062 Aachen, Germany; sduesterhoeft@ukaachen.de; 3Institute of Virology, Faculty of Veterinary Medicine, Justus-Liebig-University Giessen, Schubertstrasse 81, 35392 Giessen, Germany; benjamin.j.lamp@vetmed.uni-giessen.de; 4CIRI—Centre International de Recherche en Infectiologie, University Lyon, Université Claude Bernard Lyon 1, Inserm, U1111, CNRS, UMR5308, ENS Lyon, 46 Allée d’Italie, 69007 Lyon, France

**Keywords:** pestivirus, ADAM17, maturation, receptor downregulation

## Abstract

Upon host cell infection, viruses modulate their host cells to better suit their needs, including the downregulation of virus entry receptors. ADAM17, a cell surface sheddase, is an essential factor for infection of bovine cells with several pestiviruses. To assess the effect of pestivirus infection on ADAM17, the amounts of cellular ADAM17 and its presence at the cell surface were determined. Mature ADAM17 levels were reduced upon infection with a cytopathic pestivirus bovis (bovine viral diarrhea virus, cpBVDV), pestivirus suis (classical swine fever virus, CSFV) or pestivirus giraffae (strain giraffe), but not negatively affected by pestivirus L (Linda virus, LindaV). A comparable reduction of ADAM17 surface levels, which represents the bioactive form, could be observed in the presence of E2 of BVDV and CSFV, but not LindaV or atypical porcine pestivirus (pestivirus scrofae) E2. Superinfection exclusion in BVDV infection is caused by at least two proteins, glycoprotein E2 and protease/helicase NS3. To evaluate whether the lowered ADAM17 levels could be involved in superinfection exclusion, persistently CSFV- or LindaV-infected cells were challenged with different pestiviruses. Persistently LindaV-infected cells were significantly more susceptible to cpBVDV infection than persistently CSFV-infected cells, whilst the other pestiviruses tested were not or only hardly able to infect the persistently infected cells. These results provide evidence of a pestivirus species-specific effect on ADAM17 levels and hints at the possibility of its involvement in superinfection exclusion.

## 1. Introduction

Pestiviruses (family *Flaviviridae*) are important pathogens of cloven-hooved animals. For several decades, pestiviruses were thought to solely infect cloven-hooved animals, but recent studies revealed their presence in porpoises [1,2], pangolins [3,4], bats [5,6] and rats [7,8]. Pestivirus particles harbor four structural proteins. Within the viral envelope, the core protein forms the nucleocapsid together with the single-stranded, positive-sense RNA genome of the virus [9]. Three glycoproteins are integrated into the viral envelope, E^rns^, E1 and E2. E^rns^ is involved in initial attachment events due to its interaction with cell surface glycosaminoglycans [10,11]. Apart from Bungowannah virus (*Pestivirus australiense*), it is required for the formation of infectious particles [12]. Additionally, it is involved in innate immune evasion, which is mediated by its ability to degrade RNA [13,14]. E1 and E2 are primarily present as heterodimers at the surface of the virus particles. E2 interacts with virus receptors at the cell surface. The function of E1 is not known to date, but it is likely important for fusion [15,16].

Pestivirus entry factors have mostly been studied employing either bovine viral diarrhea virus 1 (BVDV-1, *Pestivirus bovis*) or classical swine fever virus (CSFV, *Pestivirus suis*). Bovine CD46 was identified as a receptor for BVDV-1 [17], whilst porcine CD46 serves as an entry receptor for atypical porcine pestiviruses (APPV, *Pestivirus scrofae*) [18]. For CSFV, several host proteins were put forward as entry factors: laminin receptor, integrin beta 3 [19] and annexin II [20].

Recently, a disintegrin and metalloproteinase 17 (ADAM17) was identified as an essential entry factor for BVDV-1, BVDV-2 (*Pestivirus tauri*), HoBi-like pestivirus (*Pestivirus braziliense*), CSFV and LindaV (*Pestivirus L*) [21,22]. ADAM17 is an important cell surface sheddase [23]. It is responsible for the shedding of TNF-α and multiple other cell surface proteins. TNF-α is a potent mediator of inflammation. Its release, and therefore ADAM17 activity, needs to be tightly regulated to prevent pathologies associated with overstimulation of the inflammatory response. ADAM17 is synthesized as a pro-form in the ER, which relies on a close interaction with iRhoms for its trafficking from the ER to the Golgi and, subsequently, to the cell surface. Within the Golgi, ADAM17’s pro-domain is cleaved off by furine, resulting in the mature, bioactive protein.

Superinfection exclusion is the inhibition of infection of already infected cells with a related virus. It was previously described, for example, for human immunodeficiency virus (HIV) [24], feline immunodeficiency virus (FIV) [25] and hepatitis C virus (HCV) [26,27] and can be caused by receptor downregulation or inhibition of replication. For BVDV, superinfection exclusion of bovine cells was reported to depend on the E2 ectodomain and NS3 [28,29]. In line with the phenomenon of receptor downregulation, it stands to reason that superinfection exclusion of BVDV might in part be due to downregulation of the recently discovered essential entry factor ADAM17.

Here, we demonstrate that pestiviruses, and different pestiviral E2 proteins, have a species-specific impact on the amount of mature ADAM17, and ADAM17 cell surface presence. This species-specific effect correlates with the susceptibility of persistently pestivirus-infected cells to infection with a cytopathic BVDV-1 (cpBVDV-1), potentially implementing the direct role of ADAM17 in superinfection exclusion of certain pestiviruses.

## 2. Materials and Methods

### 2.1. Cells and Viruses

MDBK [30], SK6 [31], CRIB [32], HEK 293 and HEK Tet-off 293 (kindly provided by Dr. Stanislav Indik, Vetmeduni, Vienna) were cultured in DMEM high glucose in the presence of 10% pestivirus-free fetal calf serum and penicillin–streptomycin at 37 °C and 5% CO_2_. Viruses employed in this study were BVDV-1 NADL (cytopathic) [33], BVDV-1 C87 (cytopathic), CSFV 2.3 Alfort Tübingen [34], LindaV [35], BDV-1 X818 (*Pestivirus ovis*) [36], strain giraffe (*Pestivirus giraffae*) [37] and BVDV-2 890 [38]. Additionally, a previously published BVDV-1 C87 with an mCherry label at the E2 N-terminus was employed [39]. CSFV, strain giraffe and LindaV were propagated in SK6 cells, the remaining viruses in MDBK cells. MDBK and CRIB cells expressing bovine ADAM17 (MDBK_bADAM17_ and CRIB_bADAM17_) were available from a previously published study [22]. These cells express bovine ADAM17—in the case of MDBK cells, in addition to endogenous ADAM17—under the control of the EF1alpha core promoter and were generated by lentiviral transduction.

For transfection, 1.5 × 10^5^ HEK 293 or HEK 293 Tet-off cells were transfected 24 h after seeding in a 24-well plate with 300 ng plasmid preincubated with 50 µL OptiMem (Gibco, Waltham, MA, USA) and 1.25 µL 1 mg/mL polyethylenimine for 10 min at room temperature.

SK6 Tet-on cells inducibly expressing the E2 protein of APPV, CSFV or LindaV fused with mCherry at its C-terminus were generated by co-transfection of the respective pTRE plasmid and a plasmid encoding for a puromycin resistance. Cells were clonally selected by the addition of 2 µg/mL puromycin and selected clones were tested for the expression of E2-mCherry after treatment with 2.5 µg/mL of doxycycline by fluorescence microscopy.

### 2.2. Quantification of ADAM17 Levels by Western Blot

For the quantification of ADAM17 levels after infection with pestiviruses, 2 × 10^5^ MDBK cells were infected with an MOI of 1 with either BVDV-1 C87, BVDV-2 890, CSFV Alfort Tübingen, strain giraffe or LindaV. Cells were lysed 24, 48 or 72 h after infection with 250 µL protein loading buffer (6 M urea, 2% SDS, 10% glycerin, 62.6 mM Tris pH 6.8). For the quantification of ADAM17 levels in SK6 Tet-on cells expressing APPV, CSFV or LindaV E2, 1 × 10^5^ cells/well were seeded in a 24-well plate. After 24 h, E2 expression was induced by the addition of 2.5 µg/mL doxycycline in one well per cell line, whilst the other well was mock treated. After an additional 24 h, the cells were lysed with 250 µL protein loading buffer. The lysates were incubated at 95 °C for 5 min and subsequently sheared through 27 G needles. Amounts of 10 µL lysate per sample were loaded onto 7.5% Tris-Tricin gels and subsequently blotted with the TransBlot Turbo System (BioRad, Hercules, CA, USA) on low fluorescent PVDF membranes (BioRad, Hercules, CA, USA) and blocked for 5 min at room temperature with EveryBlot Blocking buffer (BioRad, Hercules, CA, USA). The blots were subsequently incubated for 60 min with a rabbit anti-ADAM17 polyclonal serum detecting the ADAM17 cytoplasmic tail (ab39162, Abcam, Cambridge, UK, 1:1000) and an anti-beta-actin mouse monoclonal antibody (Sigma, Burlington, VT, USA, 1:15,000) in TBS 0.1%Tween. After three consecutive washing steps, the blots were incubated with StarBright 700 goat anti-rabbit (BioRad, Hercules, CA, USA) and IRDye 800 CW goat anti-mouse IgG (Licor, Lincoln, NE, USA) 1:5000 for 60 min. After three consecutive washing steps, the signal was resolved in a ChemiDoc MP imager (Biorad, Hercules, CA, USA) and the signal intensities were quantified with Fiji [40]. To account for potential differences in protein amounts loaded on the gel, the ADAM17 signals were normalized to the actin signal determined on the same blot.

### 2.3. Quantification of Cell Surface ADAM17 Levels by Flow Cytometry

For the quantification of ADAM17 cell surface levels, 1.5 × 10^5^ HEK 293 cells were either transfected with a plasmid encoding for BVDV proteins Npro to E2, with an N-terminal mClover tag in E2 or the same plasmid with a deletion of the E2 ectodomain. Additionally, HEK 293 Tet-off cells were transfected with doxycycline responsive plasmids (pTRE) encoding the E2 proteins of APPV, LindaV, BVDV or CSFV with a C-terminal mCherry tag. Cells were detached 24 and 48 h after transfection by incubation with PBS for 5 min at 37 °C, fixed with 4% PFA in PBS (20 min, 4 °C) and subsequently incubated with a mouse anti-human ADAM17 ectodomain monoclonal antibody (MAB9301-100, Biotechne, Minneapolis, MN, USA, 1:50) for 45 min at 4 °C on a turning wheel. After three washing steps with PBS, the cell pellets were resuspended in a PBS solution containing an Alexa647 goat anti-mouse polyclonal serum (Invitrogen, Waltham, MA, USA, 1:300) for 45 min on a turning wheel at 4 °C before being washed three times with PBS and being resuspended in PBS + 2% FCS for subsequent analysis in an Amnis FlowSight (Luminex, Austin, TX, USA). Data analysis was performed employing the corporate IDEAS 6.3 software. Cells were gated for single cells based on diameter and aspect ratio. Subsequently, the Alexa647 surface signal intensity was quantified and compared between E2-expressing and not-expressing cells. Histograms depicted were generated with FlowJo10.

### 2.4. Quantification of Superinfection Exclusion

MDBK, MDBK_bADAM17_ and CRIB_bADAM17_ cells persistently infected with either CSFV or LindaV were generated. Freshly thawed cells were passaged twice before infection with CSFV (MOI 0.1) or LindaV (MOI 0.01). The number of infected cells was assessed after each passaging by fluorescence microscopy employing the mouse monoclonal anti-E2 antibody 6A5 [41] or the mAb A18 for CSFV [42] and a goat anti-mouse Cy3 labeled polyclonal serum (Jackson Immunologicals). Once no uninfected cells were detectable anymore for three consecutive passages, they were considered persistently infected. The presence of no uninfected cells was controlled each time these cells were employed in an experiment. To quantify the effect of superinfection exclusion, 1 × 10^5^ MDBK, MDBK_ADAM17_ or CRIB_ADAM17_ cells with or without persisting infection were seeded in each plate of a 24-well plate. The next day, they were infected with serial dilutions of different pestiviruses (BVDV-1 C87 with or without fluorophore label at E2, BVDV-1 NADL, CSFV Alfort Tübingen, LindaV labeled with a fluorophore at E2 and BDV X818). The medium was exchanged to a medium containing 1% carboxymethlcellulose 4 h after infection and the number of infected foci was determined 48–72 h after infection (depending on the individual virus). For this purpose, cells were fixed with 4% paraformaldehyde for 20 min at 4 °C. In the case of viruses encoding a fluorophore, no further staining was necessary. For the remaining viruses, a mouse monoclonal antibody (mAb) not interacting with the virus already present in the cells was chosen (mAb A18 for CSFV [42] and mAb code4 for BVDV and BDV [33]).

### 2.5. Statistical Analysis

Statistical analysis, either as a Student’s *t*-test or two-way ANOVA, was performed in RStudio 2022.07.2 Build 576 [43]. Data were visualized with Graph Pad Prism 10. At least three independent experiments were performed for each experimental setup.

## 3. Results

### 3.1. BVDV-1, CSFV and Strain Giraffe Infections Decrease Mature ADAM17 Levels

Viruses can impact the availability of their host cell entry factors. To determine if a phenomenon similar to receptor downregulation could be observed for ADAM17 upon pestivirus infection, MDBK cells were infected with different pestivirus species, cytopathic BVDV-1 (cpBVDV-1), BVDV-2, CSFV, strain giraffe (Pestivirus giraffae) or LindaV, or mock treated. Cells were lysed at 24, 48 and 72 h post-infection (hpi), and the amounts of mature and immature ADAM17 were relatively quantified by Western blot analysis compared with mock-treated cells employing an antibody interacting with the ADAM17 cytoplasmic domain (Figure 1). We observed different changes in the cellular levels of mature and immature ADAM17, as well as changes in their ratios, depending on the pestivirus species used for infection. For all pestiviruses except LindaV, mature ADAM17 levels were decreased 72 hpi by 39–83%. For all viruses except cpBVDV-1 and BVDV-2, the levels of immature ADAM17 were either equal to or higher than those in mock-infected cells.

The immature form of ADAM17 is located within the ER and not biologically active. To further examine the cellular ADAM17 pool, we calculated the relative amount of mature ADAM17 as a percentage of immature ADAM17 to estimate the balance between ADAM17 production and maturation. The percentage of mature ADAM17 increased by 21–83% for all viruses tested at 24 hpi, but then decreased by 17–46% at 48 hpi and decreased by 20–74% at 72 hpi (cpBVDV-1: 74%, *p* = 0.008; CSFV: 56%, *p* = 0.036; strain giraffe: 51%, *p* = 0.025). These results demonstrate a relative decline in mature ADAM17 levels and a relative increase in immature ADAM17 in the total cellular ADAM17 pool upon infection with cpBVDV-1, CSFV and strain giraffe. This decline may be due to decreased maturation or increased degradation of mature ADAM17.

### 3.2. Pestivirus Structural Proteins Can Affect ADAM17 Surface Levels

Mature ADAM17 is stored in the Golgi and adjacent vesicles, and its amount on the cell surface is tightly regulated and generally limited [44,45]. ADAM17 at the cell surface represents the bioactive form, which is responsible for ectodomain shedding, but also for the interaction with CSFV E2 [21]. However, reliable serological reagents to detect the ADAM17 ectodomain are only available for human ADAM17, preventing analysis of its surface levels by flow cytometry in pestivirus-infected bovine cells. To overcome this problem, we expressed cpBVDV-1 proteins Npro to E2 (BVDVNpro-E2_fluo_), with E2 carrying an N-terminal mClover tag, via plasmid DNA transfection in HEK293 cells, which resulted in a mixed population of transfected, Npro-E2-expressing cells and non-transfected cells. The presence of a fluorophore labeled E2 allowed us to easily compare the ADAM17 surface levels of BVDVNpro-E2_fluo_-expressing cells and non-transfected cells present in the same sample by flow cytometry. Expression of BVDVNpro-E2_fluo_ decreased ADAM17 surface levels as expressed in mean fluorescence intensity (MFI) when compared with non-expressing cells by 25% (*p* < 0.0001, *n* = 5) 48 h after transfection (Figure 2A, Appendix A). These results provide initial evidence for a correlation of total mature ADAM17 levels observed in pestivirus-infected cells and the presence of ADAM17 at the cell surface.

Previous studies have shown that the expression of the BVDV E2 ectodomain results in superinfection exclusion [29]. The glycoprotein E2 is resident in the ER and therefore could interact with the ectodomain of immature ADAM17 in the ER lumen. To test whether the E2 ectodomain was involved in the reduction of ADAM17 cell surface levels, HEK293 cells were transfected with an E2 ectodomain-deleted BVDVNpro-E2_fluo_ (BVDVNpro-E2_fluo_∆E2ectodomain). No reduction of ADAM17 surface levels was detectable in cells expressing BVDVNpro-E2_fluo_∆E2ectodomain 48h after transfection compared with non-expressing cells (Figure 2A). These results demonstrate that the expression of BVDV structural proteins lower ADAM17 surface levels and that this effect is likely mediated by the E2 protein.

To assess whether E2 was solely responsible for reduced ADAM17 surface levels and whether this effect was consistent across different pestiviruses, we generated constructs inducibly expressing the E2 of BVDV-1, CSFV, LindaV or APPV with a C-terminal mCherry tag. These viruses were chosen to represent two viruses lowering mature ADAM17 levels (BVDV and CSFV), one virus not showing this effect (LindaV) and one pestivirus possessing an atypical E2 (APPV). HEK293 Tet-off cells were transfected with the respective constructs and the fluorescence intensities of ADAM17 at the cell surface were determined 48 h later by flow cytometry. The MFI was calculated by comparing the fluorescence intensities of transfected and non-transfected cells present in the same well, which could easily be distinguished in the flow cytometer due to the presence or absence of an mCherry signal. A significant reduction in the ADAM17 signal at the cell surface, by 20% for BVDV E2 and 15% for CSFV E2 was observed (Figure 2B), whilst no significant changes were noted upon the expression of LindaV or APPV E2. These results demonstrate that the E2 proteins of BVDV and CSFV reduce ADAM17 cell surface levels, which aligns with the reduced amounts of mature ADAM17 observed in infected MDBK cells. In contrast, LindaV and APPV E2 do not appear to impact ADAM17 cell surface levels.

To verify that the expression of CSFV E2 was also related to lowered levels of mature ADAM17, we took advantage of SK6 cell lines inducibly expressing the mCherry labeled E2 of either APPV, CSFV or LindaV. Cells were lysed 48 h after induction of E2 expression and ADAM17 forms were quantified by Western blot. The induction of expression of APPV or LindaV E2 appeared to increase mature ADAM17 levels in comparison with untreated cells, whilst a significant decrease of more than 60% was observed upon CSFV E2 expression (Figure 2C). No such effect was detectable for the levels of immature ADAM17.

### 3.3. Superinfection Exclusion Evoked by CSFV Is Superior to LindaV

Superinfection exclusion is a phenomenon previously described for BVDV [28,29] and CSFV [46,47]. It is considered to be caused by two inhibition steps: one mediated by the E2 protein at virus entry level, and the other mediated by the NS3 protein during replication. Given that BVDV and CSFV reduced the levels of mature ADAM17, as well as cell surface ADAM17 levels, we hypothesized that downregulation of ADAM17 might play a role in superinfection exclusion. To test this hypothesis, we utilized two approaches. To check whether the cellular expression level of ADAM17 had an effect on the strength of superinfection exclusion, we used MDBK cells, MDBK cells expressing endogenous bovine ADAM17 and additional bovine ADAM17 under the control of an EF1alpha promoter (MDBK_bADAM17_) and CRIB cells, which lack endogenous ADAM17 expression and express bovine ADAM17 under the control of an EF1alpha promoter (CRIB_bADAM17_). These cell lines express different amounts of ADAM17 (Appendix A). CRIB_bADAM17_ cells harbor 75% less mature ADAM17 and 68% less immature ADAM17 compared with MDBK cells. MDBK_bADAM17_ cells contain, on average, 24% less mature ADAM17 than MDBK cells, but they harbor 142% more immature ADAM17 than MDBK cells.

To check whether the ability of the persistently infecting virus to modulate ADAM17 levels provokes an effect, MDBK, MDBK_bADAM17_ and CRIB_bADAM17_ cells persistently infected with either CSFV (decrease in mature ADAM17) or LindaV (no effect on mature ADAM17 observed) were generated. Persistently LindaV-infected cells were superinfected with serial dilutions of an mCherry labeled cpBVDV-1, BDV, CSFV or an mCherry labeled LindaV, whilst persistently CSFV-infected cells were superinfected with mCherry labeled cpBVDV-1 or mCherry labeled LindaV. Subsequently, the titres achieved on the different cell lines were determined and compared. All non-cp viruses employed in this experiment were not or hardly (titer < 10^3^ ffu/mL) able to superinfect persistently CSFV- or LindaV-infected cells, and this observation was independent of the cell line used (Appendix A), resulting in a reduction of susceptibility of more than 800 times. In contrast, cpBVDV-1 was able to superinfect both CSFV and LindaV persistently infected cells (Figure 3). Therefore, the already reported role of NS3 is reflected in our data. CpBVDV-1 titers were however significantly reduced in the persistently infected cells when compared with non-infected cells. The cellular ADAM17 levels did not significantly affect achievable titers, supporting the conclusion that either small amounts of ADAM17 are likely sufficient to mediate infection or that ADAM17 cell surface levels don’t directly correlate with the total cellular ADAM17 pool.

To assess whether the persistently infecting virus and its ability to modulate ADAM17 levels would affect the achievable titers, we compared the susceptibility of uninfected cells of cpBVDV-1 with cells infected with either CSFV or LindaV. A significant effect of the persistently infecting virus on titer reduction could be observed, with titers being on average 10-times lower for CSFV persistently infected cells.

## 4. Discussion

Viruses utilize various mechanisms to manipulate their host cells, among which are the downregulation of virus receptors and superinfection exclusion. Here, we present evidence suggesting a pestivirus species-specific effect on the levels of the essential entry factor ADAM17 and its potential implication in superinfection exclusion.

In our analysis of cellular ADAM17 levels, a significant reduction of relative mature ADAM17 levels 72 hpi could be detected in cells infected with cpBVDV-1, CSFV and strain giraffe. The downregulation of ADAM17 is difficult to assess because trafficking and maturation is complex. In the ER, ADAM17 is present in its immature or pro-form, which is enzymatically inactive. The tight interaction with iRhoms subsequently results in its trafficking to the Golgi and, afterwards, to the cell surface [23]. Within the Golgi compartment, ADAM17 undergoes maturation by furin cleavage, leading to the release of the ±23 kD pro-domain. The observed relative and absolute decrease in mature ADAM17 levels could be caused by inhibited trafficking of ADAM17 from the ER to the Golgi, impaired furin cleavage or increased degradation of mature ADAM17. It was previously demonstrated that the pro-form of ADAM17 can increase its molecular weight when cleavage of the prodomain is inhibited, most likely due to sialylation [48]. However, we did not observe a molecular weight increase in immature ADAM17 and therefore consider a ADAM17–E2 interaction in the ER the most likely mode of action. Further analyses are needed to test this hypothesis.

Superinfection exclusion has been described for several viruses (as for example reported in [24,25,26,28,29,49,50,51,52,53]), but the exact benefit for the virus remains debated. For viruses of eukaryotes, superinfection exclusion may offer several benefits, including preservation of genetic stability of a virus population [54], avoiding competition for host resources and preventing anchoring of budding/released viruses at the surface of already infected cells. Superinfection exclusion by BVDV has been shown to depend on the major glycoprotein E2 and the protease/helicase NS3, with E2 acting at the level of virus entry and NS3 at the level of replication [28,29]. For the E2-effected virus entry block, a time-dependent phenomenon was described, with a complete loss of susceptibility for a cp-BVDV strain of cells infected for 12 h with a non-cp BVDV strain, which was reduced to an approximately ten-fold reduction 3 days after infection, which persisted upon the passaging of the cells. Therefore, it may be speculated that NS3-independent superinfection exclusion might be caused by two potentially independent mechanisms, e.g., a high-level, short-term inhibition and a low-level, long-term inhibition.

The reduction of mature ADAM17 levels in cells infected with cpBVDV-1, strain giraffe or CSFV led us to hypothesize that the E2 ectodomain might be causing a loss of mature ADAM17. As the biologically relevant fraction of ADAM17 is located at the cell surface, we quantified this fraction by flow cytometry, employing an antibody targeting the ADAM17 ectodomain. Indeed, the expression of BVDVNpro-E2_fluo_ in HEK293 cells decreased ADAM17 surface levels by 25%, whilst no change of ADAM17 surface levels compared with mock-treated cells was observed in the absence of the E2 ectodomain (BVDVNpro-E2_fluo_∆E2ectodomain). Whilst a 25% loss of ADAM17 surface presence might be regarded as a minor change, it is important to note that knock-out of FRMD8, a factor crucial for ADAM17 localization and stabilization, reduces ADAM17 surface levels by approximately 50% in the presence of PMA [55], a compound commonly used to temporarily increase ADAM17 surface levels. Therefore, the reduction induced by the expression of BVDVNpro-E2_fluo_ is likely biologically relevant.

To determine if E2 alone could reduce ADAM17 cell surface levels and if this effect was consistent across different species, we expressed APPV, cpBVDV-1, CSFV and LindaV E2 with a C-terminal mCherry tag in HEK293 cells. Expression of cpBVDV-1 or CSFV E2 resulted in a 20% or 15% reduction in ADAM17 cell surface levels, respectively. These reductions were slightly lower than those observed for the construct expressing all BVDV structural proteins, suggesting a potential minor role of the structural proteins or a more native membrane topology of E2 in the presence of all structural proteins. LindaV E2 did not affect ADAM17 surface levels. This finding is puzzling, as LindaV also depends on ADAM17 for infection [22]. Additionally, its E2 structure is likely very similar to BVDV, as suggested by the recently published structure of the phylogenetically distant Norway rat pestivirus E2 protein [56]. Similarly, APPV E2 did not affect ADAM17 surface levels. Given its structural difference from classical pestiviruses and the lack of data concerning its dependence on ADAM17, these results seem consistent.

The differing effect of CSFV and LindaV on ADAM17 prompted us to investigate its potential role in superinfection exclusion. Non-cytopathic viruses were not or hardly able to infect persistently CSFV- or LindaV-infected MDBK cells, demonstrating the high efficiency and species independence of this process, as previously demonstrated by [46]. However, this only allowed for comparison of the infectivity between CSFV and LindaV persistently infected cells for cpBVDV-1. In this context, an average ten-fold reduction of infectivity could be observed for persistently CSFV-infected cells compared with persistently LindaV-infected cells. Considering the negative effect of CSFV E2 on mature ADAM17 availability and the lack of this effect for LindaV E2, this provides preliminary evidence for the involvement of ADAM17 in superinfection exclusion. However, the exact inhibitory mechanism and definitive proof still need to be established. Interestingly, the different ADAM17 expression levels in bovine ADAM17-expressing MDBK and CRIB cells had only a minor effect on the susceptibility of persistently infected cells. Based on this phenotype, we hypothesize that cell surface ADAM17 levels between these cell types are comparable, as only a small amount of the total cellular ADAM17 pool is usually present at the cell surface. Nevertheless, there was a trend suggesting that higher cellular ADAM17 levels attenuate the superinfection exclusion effect, with persistently infected MDBK_bADAM17_ cells being most permissive and persistently infected CRIB_bADAM17_ cells being least permissive.

Further studies are clearly warranted to comprehensively assess the intricate interplay between pestiviruses and ADAM17.

## Figures and Tables

**Figure 1 viruses-16-01564-f001:**
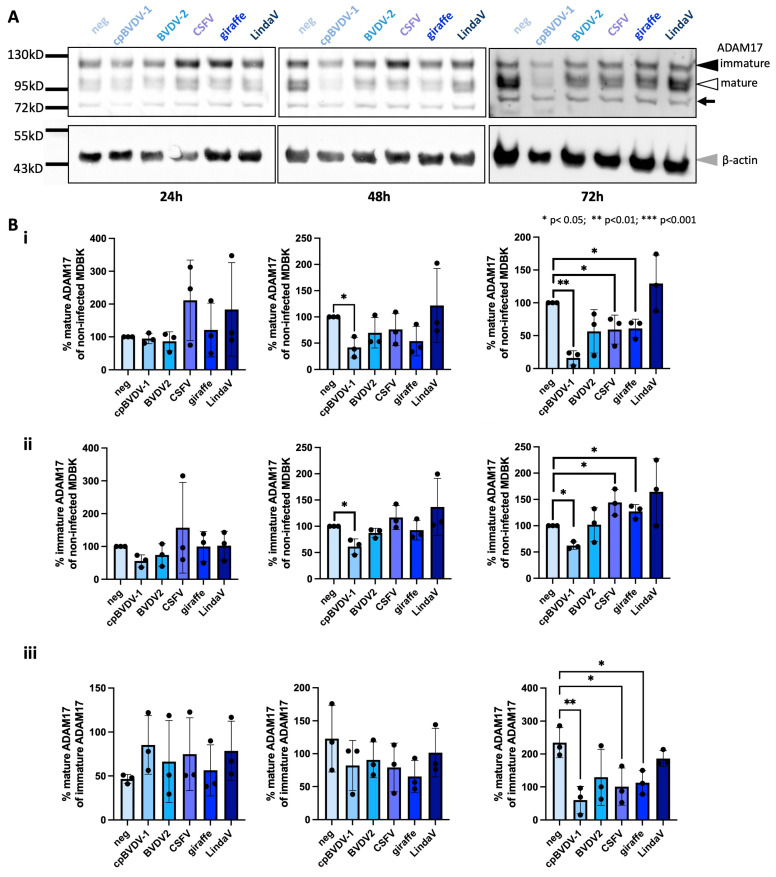
Pestivirus infection can change the cellular amounts of mature and immature ADAM17. (**A**) MDBK cells were infected with cpBVDV-1, BVDV-2, CSFV, strain giraffe or LindaV or mock infected (neg). Cells were lysed 24, 48 or 72 hpi and mature ADAM17 (white arrowhead, apparent molecular weight 95–100 kD), immature ADAM17 (black arrowhead, apparent molecular weight 120 kD) and β-actin (grey arrowhead) were detected by Western blot analysis using fluorescence-based detection. An unspecific band also detected by the employed antibody is indicated by a black arrow. (**B**) Quantification of the relative amounts (normalized to β-actin) of mature (**i**) and immature (**ii**) ADAM17 in comparison with non-infected cells is shown, as well as the ratio of mature to immature ADAM17 expressed in % (**iii**) as determined by Western blot analysis. Timepoints correspond to the timepoints given in (**A**). Shown are mean and standard deviation of *n* = 3 independent experiments. Statistical significance of the results was calculated with a Student’s *t*-test.

**Figure 2 viruses-16-01564-f002:**
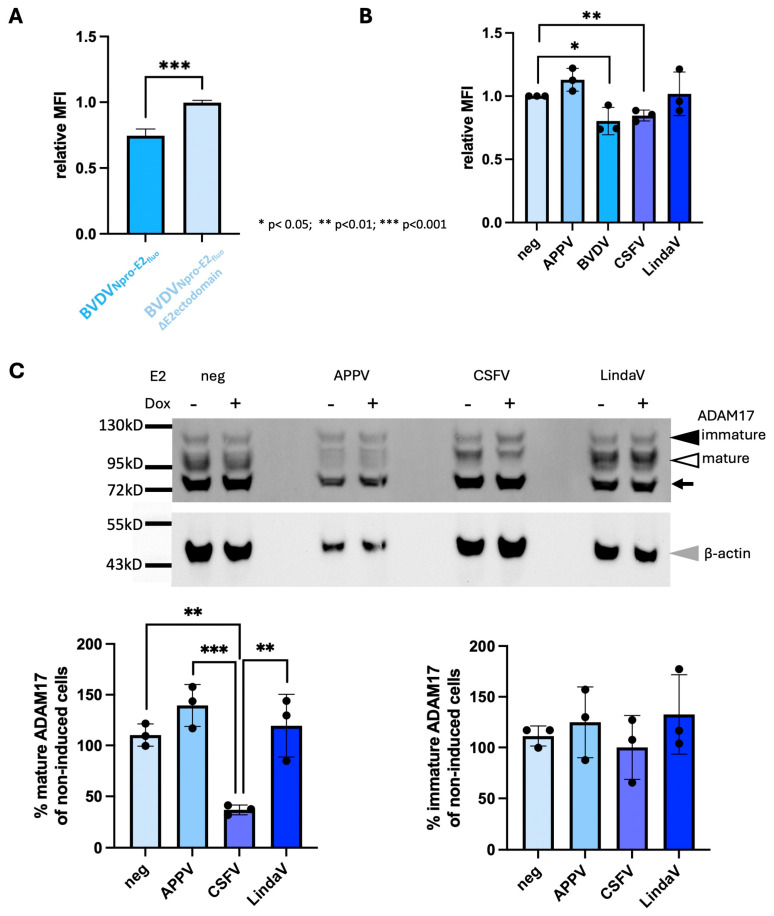
BVDV and CSFV E2 lower ADAM17 surface levels. (**A**) HEK293 cells 48 h after transfection expressing either BVDV Npro to E2 labeled at its N-terminus with a fluorophore (BVDVNpro-E2_fluo_) or the same construct with a deletion of the E2 ectodomain (BVDVNpro-E2_fluo_∆E2ectodomain) were analyzed by flow cytometry for the expression of ADAM17 at the cell surface. Relative mean fluorescence intensities were calculated by comparing the levels of surface fluorescence intensities of transfected and non-transfected cells. Shown are mean and standard deviation of *n* = 5 independent experiments. Statistical significance of the results was calculated with a Student’s *t*-test. (**B**) HEK293 Tet-off cells 48 h after transfection expressing either BVDV, APPV, CSFV or LindaV E2 with a C-terminal mCherry tag were analyzed for the surface expression of ADAM17 by flow cytometry. Relative mean fluorescence intensities were calculated by comparing the levels of surface fluorescence intensities of transfected and non-transfected cells. Shown are mean and standard deviation of *n* = 3 independent experiments. Statistical significance of the results was calculated with a Student’s *t*-test. (**C**) Changes in ADAM17 levels upon induction of pestivirus E2 expression in SK6 Tet-on cells are shown. Shown are one representative Western blot and the quantification of the % of mature and immature ADAM17 in doxycycline-induced versus non-induced cells (normalized to β-actin). Mature ADAM17 (white arrowhead, apparent molecular weight 95–100 kD), immature ADAM17 (black arrowhead, apparent molecular weight 120 kD) and β-actin (grey arrowhead) were detected by Western blot analysis using fluorescence-based detection. An unspecific band also detected by the employed antibody is indicated by a black arrow. Mean and standard deviation of *n* = 3 independent experiments are depicted. Statistical significance of the results was calculated with a Student’s *t*-test.

**Figure 3 viruses-16-01564-f003:**
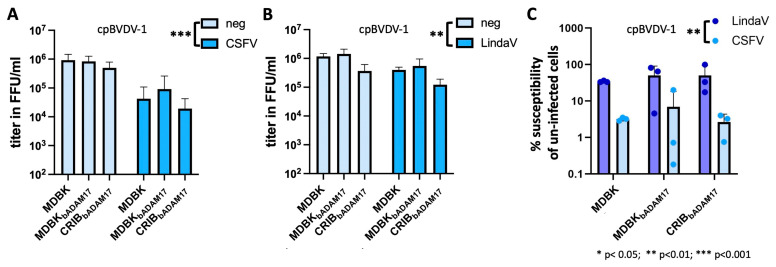
Persistently LindaV-infected cells are more susceptible to superinfection by a cpBVDV-1 than persistently CSFV-infected cells. Virus titers (blue bars) observed in MDBK, MDBK_bADAM17_ and CRIB_bADAM17_ cells persistently infected with CSFV (**A**) or LindaV (**B**) upon superinfection with cpBVDV-1 compared with titers in non-infected cells are shown. (**C**) Comparison of the percent reduction of titer of the superinfecting virus in persistently CSFV- or LindaV-infected cells compared with non-infected cells is shown. Data are presented as the mean and standard deviation of three independent experiments (*n* = 3). Statistical significance was determined using two-way ANOVA.

## Data Availability

Data are available from the authors upon request.

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
