# Peer review of "Modulation of ADAM17 Levels by Pestiviruses Is Species-Specific"

_viruses, 2024, doi:10.3390/v16101564_

Round 1
Reviewer 1 Report (New Reviewer)
Comments and Suggestions for Authors
ADAM17 is a recently discovered factor that is essential für infection of bovine cells with a variety of pestiviruses. In this project, Chen et al. show that cell surface ADAM17 is reduced upon infection with cpBVDV, CSFV, or giraffe pestivirus, but not by Linda virus. Accordingly, this reduction correlated with reduction by the viral envelope glycoprotein E2, again with Linda – and additionally, APPV – E2 being inactive. Finally, the authors provide some evidence that the reduced level of cell surface ADAM17 in CSFV, but not Linda virus, persistently infected animals might be involved in the E2-dependent mechanism of superinfection exclusion. The latter point of superinfection, however, is difficult to reconcile with the data presented. Thus, (i) cpBVDV titers were reduced on both, CSFV and Linda Virus persistently infected cells (albeit more pronounced om CSFV-infected cells) despite Linda virus not reported to alter the level of active, surface exposed ADAM17 (but Linda virus infection still dependent on ADAM17 !?), and (ii) the different levels of ADAM17 in wt MDBK, MDBK-bADAM17 and CRIB-bADAM17 does not seem to have a dose-dependent effect. Thus, it is a very interesting study that is well written and presented, but the title, even if written as a question, leads to the least convincing data.
Specific comments:
1) The conclusion stated in the abstract (“These results provide evidence of a pestivirus species specific effect on ADAM17 levels and its potential involvement in superinfection exclusion”) appears to be much stronger than the data presented and even than discussed in the manuscript. It might be worthwhile to consider tuning down the abstract and to include the regulation of the ADAM17 level by pestivirus infection in the title and not to focus exclusively on the effect on superinfection exclusion, despite agreeable, the current title is pretty catchy.
2) E.g., in the Methods section, it should be stated which virus strain used is cytopathogenic or non-cytopathogenic. It might not be evident for non-pestivirus afficionados.
3) For the reader, it’s not always evident which virus was used or not for which reason. E.g., giraffe pestivirus was used in 3.1. (quantification of mature ADAM17 level), but was not used anymore later.
Similarly, it might be explained why for some experiments, expression of Npro-E2 was used, whereas in other experiments an/or with other viruses, tagged-E2 expression was used exclusively.
In addition, can it be excluded that a different species of pestiviruses are more or less dependent on ADAM17, e.g., using GAG-binding more efficiently, as was reported for the giraffe strain PG-2 but not H138 (giraffe-1)?
4) Legends to Figure 1 and 2: The description of the arrows in the western blot is not correct. Immature ADAM17 seem to be the black arrowhead, the mature form the white arrowhead, not vice versa. In the legend to Supplementary Figure 2, the arrows and arrowheads are mixed even worse.
5) In the study by Lee et al. (Ref 28), the E2-based superinfection exclusion was transiently and most evident upon acute infection, but was lost after passaging the cells, i.e., in persistently infected cells. Thus, might the effect of reduction of mature ADAM17 also be transient, or at least more pronounced upon acute infection? This might be experimentally tested (but not requested here as a prerequisite for publication) but at least be discussed.
6) Discussion, lanes 344-350. The discussion on bacteriophage superinfection exclusion appears a little bit far-fetched and its relation to the current study is not really evident. Can be probably deleted without loss of information for the discussion, or otherwise, the relation to pestivirus superinfection exclusion needs to be explained.
7) Discussion, lanes 391-396. The working hypothesis for the role of ADAM17 in superinfection exclusion should be explained in more detail. Thus, the reasoning is difficult to follow with a rather limited reduction of cell surface ADAM17, with a small among of ADAM17 being sufficient to infect the cells.
8) The references need quite some editing. References were probably generated by a reference software, but the data in the database might not be complete or incorrect.
E.g. (list possibly not complete)
- Refs 2, 4, 18 (online-only journals): The article number is missing (Ref 2 have 11 pages, it’s not pp.1-11
- Refs 12, 14, 28, 36, 59: Provide full references. Volume and pages or article numbers are missing.
- Ref 43. Is there an online link available, or is this a book, book chapter??
- Refs 4, 9, 36: doi number is missing.
9) Line 290: persistently, not persistenly
Author Response
We sincerely thank this reviewer for his / her efforts. Our detailed responses to all remarks can be found in the attached pdf file.

Reviewer 2 Report (New Reviewer)
Comments and Suggestions for Authors
In the manuscript, Chen et al., described their new findings on relationship between cell surface ADAM17 levels and pestivirus superinfection. They showed that several members of the pestivirus infection lead to downregulation of the cell surface ADAM17 levels in cultured cells, and this downregulated ADAM17 could restrict superinfection of other pestivirus. Although the mechanism behind these phenomena remains elusive and there are many open questions, the finding in the manuscript is quite interesting and the manuscript is clearly written. This reviewer has no further questions on the manuscript.
Author Response
Comments 1: In the manuscript, Chen et al., described their new findings on relationship between cell surface ADAM17 levels and pestivirus superinfection. They showed that several members of the pestivirus infection lead to downregulation of the cell surface ADAM17 levels in cultured cells, and this downregulated ADAM17 could restrict superinfection of other pestivirus. Although the mechanism behind these phenomena remains elusive and there are many open questions, the finding in the manuscript is quite interesting and the manuscript is clearly written. This reviewer has no further questions on the manuscript.
Response 1: We sincerely thank this reviewer for the evaluation of our manuscript.
This manuscript is a resubmission of an earlier submission. The following is a list of the peer review reports and author responses from that submission.
Round 1
Reviewer 1 Report
Comments and Suggestions for Authors
This manuscript showed that the protein levels of mature ADAM17 in MDBK cells decreased after infection with three types of pestiviruses (BVDV, CSFV and strain giraffe), but not with Linda virus. Using flow cytometry method, the surface ADAM17 was reduced in HEK293 cells expressing the full-length E2 proteins of BVDV and CSFV. In addition, cells persistently infected with Linda virus are more susceptible to BVDV than cells persistently infected with CSFV. The data presented is solid and convincing, but some additional revision could improve the impact.
Points:
-Page 2, Line 52-53, which ref. mentioned that BDV (Pestivirus ovis) uses ADAM17 as an entry factor? I couldn’t find the data in ref 21 and 22. This information is also relevant to the Fig. 4B, where BDV was used to superinfect the persistently LindaV infected cells.
-Page 4, Line163-164, It clearly showed that the immature ADAM17 level decreased upon cpBVDV-1 infection (Fig. 1B, central panel). Please describe the results more precisely.
- For Fig.4, did the ADAM17 levels change after persistent infection with CSFV and LindaV? It would be really helpful to know whether acute or persistent infection would result in the similar changes of ADAM17.
-“whilst the other pestiviruses tested did not or hardly replicate in the persistently infected cells.” in Abstract, Page 1, Line 26-27. Was there any result supporting this conclusion? The virus titers below the detection limit could result from the entry defects, unless the authors prove virus entry was unaffected.
-Fig.4B the cpBVDV panel, better draw the significance of MDBKbADAM17 vs CRIBbADAM17 (*) above the corresponding bars, which will be clearer.
-Do not begin a sentence with numerals, instead, use words. For example, Page 3, Line143. There’s more in the Methods. Please correct throughout.
-Please add brief information about how persistent infected cells were generated in the Methods.
-The paper could benefit from detailed language editing. There are some sentences that are awkward or grammatically incorrect.
Comments on the Quality of English LanguageThe paper could benefit from detailed language editing. There are some sentences that are awkward or grammatically incorrect.
Reviewer 2 Report
Comments and Suggestions for Authors
The study by Chen et al. looks at the role of ADAM17 in superinfection exclusion in pestiviruses. The group identified that some strains of pestiviruses are able to modulate ADAM17 levels, such as cpBVDV-1 and CSFV, resulting in reduced levels of mature ADAM17, whereas others, such as LindaV, cannot. The group showed that surface mature ADAM17 reduction can be attributed to viral E2 protein, with BVDV E2 and CSFV E2 able to downregulate surface ADAM17, but not LindaV E2 or APPV E2. Using cell lines expressing different levels of ADAM17, and different strains of pestiviruses, the group then looked at superinfection exclusion. They first infected cells with either CSFV or LindaV, and then followed up with a secondary infection with a different strain of pestivirus, measuring the viral titre of the second virus. The results suggest that cells persistently infected with CSFV were less susceptible to secondary infections by BVDV, when compared to LindaV persistently infected cells. Although the authors propose that observed phenotype can be attributed to ADAM17 downregulation in CSFV infection, that was not directly tested in this study, and I have made a few comments about that below.
Overall, the authors provide interesting and important results about pestiviruses and ADAM17 biology. The data in the manuscript are mostly convincing, however some claims need to be better supported by additional data. Also, some figures/result paragraphs need to be revised to make it clearer to understand. Please see some points below, addressing of which may improve the manuscript.
Major points:
1) Have the authors performed any ADAM17 blocking/silencing experiments where they blocked ADAM17 in persistently LindaV infected cells and then infected with a second pestivirus to see if cells become more resistant to superinfection? This is a crucial experiment if the authors wish to claim that ADAM17 downregulation by pestiviruses promotes superinfection exclusion.
2) Related to the first point, have the authors looked at the role of ADAM17 in primary pestivirus infections? Their data in Figure 4 suggest that without persistent infection (labelled uninfected in the figure), even despite different expression levels of ADAM17 in the bovine ADAM17 expressing MDBK and CRIB cells, there doesn’t seem to be statistically significant differences in viral titres. This would suggest that ADAM17 is not involved in the viral entry of primary virus, however the claim is that it is involved in superinfection? Could authors please comment their thoughts on that.
3) Figure 2 – Some representative flow cytometry histogram plots for surface ADAM17 levels for discussed viruses would aid the reader. Was it the geometric mean fluorescence intensity that was used to generate this figure? I assume so, as the authors used log scale to plot their data, however it is not specified if the mean fluorescence intensity plotted was geometric or arithmetic. Even more reason to show raw histogram data of the shifts in ADAM17 levels.
Minor points:
1) Line 14, 43 - The authors say “virus receptors”, the wording of which implies the presence of virally encoded proteins on the cell surface. Would the use of “viral entry receptors” be more accurate?
2) Lines 51-53 – It would be useful to include a sentence explaining how ADAM17 is involved in pestivirus entry. The wording here implies that it’s an entry factor, however later on the authors refer to it as a receptor on a couple of occasions (Lines 64, 140). Calling it a ‘receptor’ would imply that ADAM17 is directly involved in virus uptake, yet the two papers referenced at that statement [21, 22] show that ADAM17 is a binding factor of E2. The authors need to be careful with wording and call ADAM17 entry/binding factor, rather than a receptor.
3) Figure 1 (A) – The 72 h Western blot looks a little bit too overexposed, as demonstrated by the bleaching of signal in the 2nd and 3rd lanes. The authors may want to reduce the contrast of the image slightly, or put in a different exemplar blot altogether.
4) Results 3.1 – More information about the different ADAM17 bands on the blots should be provided, instead of simply stating that one is mature and one is immature, i.e. either provide information on the sizes of mature and immature forms of ADAM17 or provide blots where EndoH/PNGaseF digests were performed to validate the maturation status of bands (if available).
5) Results 3.1 – This whole results paragraph is very heavy with percentage data, and it’s quite difficult to follow. Have the authors considered plotting this data as a fold change instead of percentage change? It will not change the data but might be easier to interpret in text using fold change.
6) Related to Figure 2, authors don’t mention what anti-ADAM17 antibody they used for their flow cytometry staining. The methods section says “mouse anti human ADAM17 ectodomain monoclonal antibody”, but doesn’t give information of its name/manufacturer/clone/source.
7) Figure 3 (B) – There’s no key, so it is impossible to know what each bar means. Based on the colours, it’s MDBK for blue, MDBKbADAM17 for dark blue and CRIBbADAM17 for light blue, but it doesn’t make sense as the data is plotted as %ADAM17 of MDBK.
8) Lines 255-256 – How was cellular susceptibility measured? Also, is it even something that is needed in order to interpret the data? Again, having a lot of percentage data in section 3.3 of Results is hard to follow. Wouldn’t it be easier to present viral titres and discuss the differences in titres as fold changes?
9) Figure 4 – The Y scale should be presented in a scientific format, i.e. 104 instead of 10000.
10) Lines 309-311 - Is there a reference for the statement that immature ADAM17 can increase its molecular weight when cleavage of the prodomain is inhibited?
11) Lines 316-321 – The authors need to be careful with the statement that pestiviruses downregulate ADAM17 to avoid inflammatory responses by inhibiting TNFα and other cytokine cleaving. Although, the inhibition of cytokine cleaving would most likely benefit the virus, ADAM17 is also known to cleave TNF Receptors 1 and 2. If ADAM17 is downregulated by pestiviruses, the levels of surface TNFR1 and TNFR2 would go up, which could in turn increase TNFα-mediated cytokine production. Such increases in TNFR1 and TNFR2 have been shown during human cytomegalovirus infection following ADAM17 targeting (Rubina, et al, PNAS 2023).
Comments on the Quality of English Language
Overall, the quality of English is good in this manuscript. Some paragraphs are difficult to understand, but that might be due to the nature of the results, i.e a lot of percentage data.
Reviewer 3 Report
Comments and Suggestions for Authors
This study concentrated on the correlation between ADAM17 levels and pestivirus, however, here are some problems of this study:
1. The image quality of immunoblots in Figure 1A is too bad, besides, the specific antibody needs to be marked in the figure 1A and figure 3A.
2. Data in Figure 1B seems with bad repeatability.
3. The data of flow cytometry in Figure 2 need to be added.
4. Independent and explicit legend needs to be added in figure 3B.
5. The figures are disordered and these data are far from enough to support the conclusion of this study.
Comments on the Quality of English LanguageThe language needs to be edited and the scientific quality of this manuscript is poor, which needs a profound revision.
Round 2
Reviewer 2 Report
Comments and Suggestions for Authors
Relating to major points 1) and 2) from the first review round, the authors state in their reply that pestivirus infection is completely dependent on ADAM17, supporting their statement by peer reviewed references. However, their data from Figure 4 where cells with different ADAM17 levels were infected with different pestiviruses show virtually no difference in virus titres in primary infection (apart from panel B cpBVDV-1, where the viral titres in CRIBbADAM17 cells are significantly lower than MDBK bADAM17). Can the authors provide the explanation for that? I.e. viral titres in CRIBbADAM17 cells are virtually the same as those of MDBK in primary infection. However, according to Figure 3, CRIBbADAM17 cells have a lot less mature and immature ADAM17, compared to MDBK cells. Since ADAM17 is an essential factor for pestivirus entry, as backed by other research, wouldn’t the authors expect significantly lower viral titres in CRIBbADAM17 cells compared to the other two cell types used, due to reduced ADAM17 availability in those cells?
The lack of bovine specific blocking reagents is indeed a limitation, however, have the authors considered engineering a siRNA against bovine ADAM17 and depleting ADAM17 that way? I.e. infect MDBK with a persistent infection, then culture infected cells with siRNA against ADAM17 or siRNA control, then superinfect with another pestivirus. Although this method might require more optimisation and troubleshooting, it would be a much cleaner system, enabling the authors to draw solid and sound conclusions about the role of ADAM17 in superinfection.
Although it would be a better designed experiment, and would improve the manuscript substantially, I agree with the authors that for the point they are trying to make it is not essential to do ADAM17 blocking/silencing.
However, since the authors presented raw histograms in Figure 2 showing shifts in ADAM17 expression following overexpression of different E2 proteins in HEK293 cells, their argument that E2 modulates ADAM17 levels in pestivirus expression is not backed by sound data. The overall ADAM17 shifts across all examples are not convincing. It looks like the parental cell line used to overexpress different E2 proteins doesn’t have any endogenous surface ADAM17 (or just has the smallest amount in some histograms), hence expressing E2 proteins in this cell line to study its role in ADAM17 impairment doesn’t seem like the best option. A cell line with high surface ADAM17 levels should be used to then see a convincing reduction in ADAM17 levels when E2 proteins are overexpressed. The histograms presented show the smallest shifts in ADAM17 expression to be considered significant, and are not convincing enough to draw solid conclusions.
Also, representative histograms in Figure 2 are of unacceptably low quality for publication. If FlowJo is used to generate the figures, exporting in .svg format keeps quality high. There is also a lack of explanation for what “untreated control” is – is that unstained cells? If that’s unstained cells, which ones is it – the non-expressing unstained or E2 expressing unstained? Ideally, an isotype control should have been used as a negative control, instead of unstained sample (if I untreated means unstained in this instance).
Reviewer 3 Report
Comments and Suggestions for Authors
The design of this manuscript has a little flaw, besides, the figures have low quality and some of them are non-standardized. Here are some concrete problems:
1. The authors didn't replace the immunoblot images in Figure 1A, actually, the blots have low qualities and the beta-actin seems not consistent.
2. The annotation of significance in Figure 1B should be consistent. (*) p<0.1, this is very confusing and non-standardized.
3. The histogram should be depicted according to the flow cytometry figures, which could be more convincing. Authors should update the quality of figures and modify the figure legend.
4. It’s hard to visualize blots of ADAM17 in CRIBbADAM17 cells, besides, the Y-axis in Figure 3B seems unreasonable.
5. The format of references are not unified, especially the name of journals.
Comments on the Quality of English LanguageThe munuscript needs further modified to make it more concise and easy to read.